# Basic Helix-Loop-Helix Transcription Factors: Regulators for Plant Growth Development and Abiotic Stress Responses

**DOI:** 10.3390/ijms24021419

**Published:** 2023-01-11

**Authors:** Zhi-Fang Zuo, Hyo-Yeon Lee, Hong-Gyu Kang

**Affiliations:** Subtropical Horticulture Research Institute, Jeju National University, Jeju 63243, Republic of Korea

**Keywords:** bHLH transcription factor, plant growth and development, plant metabolism synthesis, plant signaling, plant abiotic stress response, crop breeding

## Abstract

Plant basic helix-loop-helix (bHLH) transcription factors are involved in many physiological processes, and they play important roles in the abiotic stress responses. The literature related to genome sequences has increased, with genome-wide studies on the bHLH transcription factors in plants. Researchers have detailed the functionally characterized bHLH transcription factors from different aspects in the model plant *Arabidopsis thaliana*, such as iron homeostasis and abiotic stresses; however, other important economic crops, such as rice, have not been summarized and highlighted. The bHLH members in the same subfamily have similar functions; therefore, unraveling their regulatory mechanisms will help us to identify and understand the roles of some of the unknown bHLH transcription factors in the same subfamily. In this review, we summarize the available knowledge on functionally characterized bHLH transcription factors according to four categories: plant growth and development; metabolism synthesis; plant signaling, and abiotic stress responses. We also highlight the roles of the bHLH transcription factors in some economic crops, especially in rice, and discuss future research directions for possible genetic applications in crop breeding.

## 1. Introduction

In all eukaryotic organisms, transcription factors regulate many biological processes by controlling the expressions of downstream target genes. Transcription factors can be grouped into different subfamilies according to their DNA-binding domains [1]. Basic helix-loop-helix (bHLH) transcription factors are among the superfamilies that are commonly found in plants and animals [2]. The conserved bHLH domain contains approximately 60 amino acids (aa), including a basic DNA binding region and two amphipathic α-helices that are separated by a loop region with a variable length [3]. The basic region consists of the first 15 amino acids. Most bHLH proteins have a glutamic acid residue at position 9 (E_9_), which can interact with the CA nucleotides in the DNA sequence [4,5]. Around 15 base-pair α-helices region in the C-terminus contain several hydrophobic amino acids, such as isoleucine (I), leucine (L), and valine (V), which promote the formation of dimeric complexes [5]. In addition, the variable loop region found in the middle of two α-helices is involved in the formation of the homodimers or heterodimers between bHLH proteins [5].

Metazoans are classified into six distinct phylogenetic groups (A–F) based on expression patterns, dimerization selectivity, and DNA-binding specificities of the bHLH domains [6]. Group A proteins bind the CAGCTG E-box configuration to form homodimers or heterodimers, such as E12, E47, and Daughterless (Da). Group B proteins bind the CACGTG E-box configuration to play important role in various developmental and cellular processes. Group C proteins contain the PAS domain that bind the ACGTG or GCGTG core sequences, which control the development of neurogenesis and the midline, as well as the formation of the tracheal and salivary ducts. Group D proteins are unable to bind DNA due to the absence of the basic domain that neutralizes the DNA-binding activities through heterodimerization. Group E proteins recognize typical sequences in the N box (CACGCG or CACGAG) that contain another two characteristic domains (‘Orange’ domain and WRPW peptide) in their C-terminus. Group F proteins contain a highly conserved COE domain that is involved in dimerization and DNA binding [7,8].

In plants, scientists have classified the bHLH transcription factors based on their characteristics. Pires and Dolan [4] studied the evolution of the bHLH transcription factors from land plants to algae, and they classified 544 bHLH transcription factors into 26 subfamilies. A total of 20 subfamilies shares the same ancestors as extant moss and vascular plants, while 6 subfamilies exist in vascular plants. In addition, Carretero-Paulet [9] present an updated and comprehensive classification of the bHLH transcription factors that extend the subgroup to 32 in plants using 638 *bHLH* genes. The plant bHLH transcription factors regulate many growth and development processes, such as embryo growth (*Retarded Growth of Embryo1, RGE1*), development of the reproductive organs including gynoecium (*HECATEs*, *HECs*) and anthers (*Dysfunctional Tapetum1*, *DYT1*), fruit dehiscence (*INDEHISCENT*, *IND*), and seed dispersal (*ALCATRAZ*, *ALC*) [10,11,12,13,14]. Furthermore, bHLH proteins also function in metabolism biosynthesis and signal transduction, such as in anthocyanin synthesis (*Transparent Testa8*, *TT8*), light signaling (*Phytochrome Interacting Factors*, *PIFs*), and brassinosteroid signaling (*BEEs*) [15,16,17]. Moreover, some bHLH subfamily members play important roles in plant biotic and abiotic stress responses, such as pathogen *Xanthomonas albilineans* (*SsbHLH15/17*), cold (*Inducer of CBF Expression1/2*, *ICE1/2*), and salt stress (*SlbHLH*) [18,19,20,21].

Due to the progressive evolution of sequencing technologies, many genome databases are available for many important plants. Recently, many researchers have performed genome-wide studies on bHLH transcription factors in various crops, and they have widely studied the functions of bHLH transcription factors in the model plant *Arabidopsis thaliana*; however, we lack extensive studies on the functions of the bHLH transcription factors in other plants. In this review, we first classify the information on the functionally characterized bHLH transcription factors according to four categories: (i) plant growth and development; (ii) metabolism synthesis; (iii) plant signaling, and (iv) abiotic stress response, and we explore their roles in various crops, and especially in rice. We also discuss the research gap and perspectives for future research for the genetic applications of bHLH transcription factors in crop breeding.

## 2. The bHLH Transcription Factor Family in Plants

Researchers first identified genome-wide *bHLH* genes in plants from *Arabidopsis* (147) and rice (167), and it has become one of the largest transcription factor families in their host plants [22,23]. Nowadays, researchers have widely identified *bHLH* gene families in various species, and they have grouped them into different numbers of subfamilies, which we present in Figure 1 (Figure 1). The bHLH protein evolutionary relationship between different species suggest that most of these subfamilies are derived from the same ancestors, and may play a fundamental role during the plant development and evolution. The functional losses of some subfamily proteins during plant evolution may have led to decreases in their evolutionary branches. In this review, we divide the bHLH transcription factors into 16 subfamilies, which we present in Table 1. Researchers have extensively studied bHLH subfamily III, which includes III(a+c), III(b), III(d+e), and IIIf; however, the *bHLH* genes in subfamilies V, VIIIb, and XI have rarely been identified, which indicates the tendency towards the bHLH transcription factors in plants.

### 2.1. bHLH Transcription Factors Are Responsible for Plant Growth and Development

bHLH transcription factors regulate plant growth and development. Researchers have demonstrated *bHLH* gene functions in *Arabidopsis*, as summarized by Hao et al. [65]. However, few *bHLH* genes have been characterized in other crop plants. Here, we focus on the functionally characterized *bHLH* genes in different plants, and mainly in crops (Figure 2).

The members of the same subfamily always have similar functions. In flowering plants, the anther is a reproductive organ that is composed of meiocytes and four cell layers: the epidermis, endothecium, middle layer, and tapetum [66]. The rice bHLH transcription factor in subfamily II, TDR Interacting Protein 2 (TIP2), plays an important role in the formation of the middle layer and tapetum during early anther development through the regulation of Tapetum Degeneration Retardation (TDR) and Eternal Tapetum 1 (EAT1). TDR and EAT1 are the key regulators in rice tapetal programmed cell death [67]. EAT1, which is a conserved bHLH transcription factor in land plants, is involved in this complex process via the direct activation of the transcriptions of two aspartic protease encoding genes (*AP25* and *AP37*) [68]. In addition, EAT1 also regulate meiotic phasiRNA biogenesis in anther tapetum, and Undeveloped Tapetum 1(UDT1) is a potential interacting partner of both EAT1 and TIP2 during the early meiosis process in rice [69,70]. Furthermore, rice *TIP2* and *EAT1* belong to subfamily II, which indicate that the *bHLH* genes in subfamily II may act as the key transcriptional regulators in anther development (Table 1). The overexpression of *OsbHLH35* endows plants with small and curved anthers, which result in a reduction in the seed production (Figure 2) [71]. In this process, three Growth Regulating Factor (GRF) family members, including OsGRF3, OsGRF4, and OsGRF11, act as the transcriptional regulators of *OsbHLH35*, and OsGRF11 acts as a negative regulator of *OsbHLH35* in rice. In other plants, *LoUDT1* from the oriental lily hybrid Siberia (*Lilium* spp.), which is the homologous gene of *OsUDT1*, is also related to anther development [72]. *Citrullus lanatus Abnormal Tapetum 1* (*ClATM1*), which is the first male sterility gene in watermelon, encodes a bHLH protein, and plays important role in the regulation of anther development, which researchers verified via CRISPR/Cas9-mediated mutagenesis [73]. In tomato, *Solyc01g081100*, which is a homolog of *OsEAT1*, is the candidate gene for the dysfunctional pollen and tapetum development in the *male sterile* 32 (*ms32*) mutant, and the CRISPR/Cas9-mediated modification of the bHLH protein encoded gene *Solyc02g079810* causes male sterility in tomato plants [74,75].

The antagonist of the PGL1 (APG) of rice in subfamily VII(a+b), called OsPIL16, controls the grain length and weight [76]. OsPIL16 interacts with two atypical bHLH transcription factors, Regulator of Grain Length 1 (PGL1) and PGL2, to antagonistically regulate the development of the rice grain length [76,77]. The overexpression of *PGL1* in lemma/palea increased the grain length and weight in transgenic rice [77]. In addition, OsPIL15 shares a close genetic relationship with OsPIL16, which also influences cell division by affecting the transport of cytokinin (CTK), which results in decreased cell numbers in rice grains [78]. Yang et al. [79] demonstrated that *OsbHLH107*, which is in the same subfamily as the *OsPIL* genes, also participates in the regulation of grain size. In another important crop plant, maize, transcription factor ZmbHLH121 in subfamily VIIIb positively regulates the kernel size and weight of maize [80].

Root hairs are long tubular projections of trichoblasts, which are the hair-forming cells on the epidermis of the plant root. They increase the plant surface area to improve absorption of nutrients from soil [81,82]. The rhizoid root hairs are essential for ion exchange, anchorage functions, and microbial interactions in the soil of land plants [81]. Various bHLH transcription factors participate in this critical root development process [83]. During root hair cell differentiation, the hair and non-hair cells are differentiated from morphologically identical epidermal cells [84]. The root hair cells produce an outside long tubule from the hair-forming cells on the epidermis of the plant root, and they function in the absorption of nutrients and water, and in interaction with microbes [82,83]. In

*Arabidopsis root hair defective 6* (*Atrhd6*) and *root hair defective six-like1* (*Atrsl1*) double mutants that lack the *RSL* class I gene function, transformation with *35S: OsRSL1*, *35S:OsRSL2* or *35S:OsRSL3* restored the expressions of the *RSL* class II genes, which indicates the functional conservation of the *RSL* genes between rice and *Arabidopsis* in root hair development [85]. In *Brachypodium distachyon*, the *RSL* class I genes, including *BdRSL1, BdRSL2,* and *BdRSL3* also promote root hair development [86]. In sweet sorghum, researchers identified a new atypical bHLH transcription factor, *SbbHLH85*, in subfamily VIIIc(2), as a key gene for root development via increase in the numbers and lengths of the root hair via ABA and auxin signaling pathways [87].

### 2.2. bHLH Transcription Factors Play Important Roles in Plant Metabolism Synthesis

Anthocyanins are the major pigments of flavonoid compounds, and they endow plants with colors, such as blue, purple, and red, in many flowers, fruits, and vegetables [88,89]. In plants, anthocyanins attract pollinators or seed dispersers, and they protect against UV radiation, pathogen attacks, and abiotic stresses [90,91]. Furthermore, anthocyanins are compounds with potential health-benefits for lowering the risk of cardiovascular diseases, certain cancers, and diabetes in humans due to high levels of antioxidant activity [92,93,94]. The MBW complex (R2R3-MYB, bHLH, and WDR) mediates the anthocyanin biosynthetic pathway, which is one of the most conserved and well-studied secondary metabolism pathways in plants [95,96].

The functionally regulated *bHLH* genes in anthocyanin biosynthesis are primarily classified as subfamily IIIf. Petroni and Tonelli [90] and Jaakola [97] reviewed the *bHLH* genes that are involved in anthocyanin biosynthesis in various horticultural species, such as petunia, antirrhinum, and grape (Table 1). In rice, Sun et al. [98] explored the minimal MBW members required for anthocyanin biosynthesis, including *S1* (*bHLH*), *C1* (*MYB*), and *WA1* (*WD40*). Under chromium stress, the rice MBW complex is regulated by the jasmonate (JA) signal and represses anthocyanin accumulation in tissues (Figure 2) [99]. Additionally, low temperature induced *SlAH* in subfamily IVd regulates anthocyanin biosynthesis in tomatoes to protect young seedlings from cold stress, which indicates the bHLH transcription factor functional connections between anthocyanin biosynthesis and abiotic stress [100]. In *Freesia hybrida*, subfamily IIIf, the bHLH transcription factors FhTT8L and FhGL3L interact with FhMYB5 in proanthocyanidin biosynthesis during flower pigmentation [101]. Peach *PpbHLH3* regulated the anthocyanin biosynthesis with *MYB10.1* and *MYB10.3* in fruit development during ripening (at the transition from the S3 to S4 stage) [102]. In sweet cherry, PabHLH3 enhances anthocyanin synthesis with PaMYB10.1-3 in fruits. This process is inhibited by PabHLH33, which is another bHLH transcription factor in subfamily IIIf [103]. The bHLH transcription factor AcB2 from onion is associated with anthocyanin accumulation via the interaction with AcMYB1, which acts as an activator in the flavonoid biosynthetic pathway in the epithelial cells of onion bulbs.

In addition, the rice bHLH transcription factor Diterpenoid Phytoalexin Factor (DPF) in subfamily IVd positively regulates the expressions of the diterpenoid phytoalexin (DP) biosynthesis genes in the process of DP accumulation. DPF was the first bHLH transcription factor to be characterized in DP biosynthesis through the N-box (5′-CACGAG-3′) [104]. In an orphan group from tomato, the bHLH transcription factor SlAR plays an important role in the carotenoid biosynthesis in the fruits, which may particularly affect lycopene accumulation [105].

### 2.3. bHLH Transcription Factors Are Involved in Plant Signaling

Green plants obtain most of their energy from light through photosynthesis, and light is an important environmental factor that determines plant growth and development. Plants can sense the red, far-infrared, and blue light spectra through photoreceptor systems, including phytochromes (PHYs), cryptochromes (CRYs), and phototropins (PHOTs), and they can then mediate the transcriptional networks in the light-regulated processes [106,107,108,109,110]. In *Arabidopsis*, members of the phytochrome-interacting factors (PIFs) in subfamily VII(a+b), such as PIF1/ PIF-like (PIL5), PIF3, PIF4, PIF5/PIL6, PIF6/PIL2, and PIF7, can interact with PHYs and play central roles in light signaling regulation [65].

Rice has six *PILs* (*OsPIL11* to *OsPIL16*) in subfamily VII(a+b), and some of the members are involved in the light signaling pathways [109,110,111,112]. *OsPIL15* was negatively regulated by light with the onset of light exposure in etiolated seedlings [113]. The overexpression of *OsPIL15* produced exhibited shorter above-ground parts, an undeveloped root system, smaller tiller angles, and enhanced shoot gravitropism, which were related to the skotomorphogenesis development, which was likely regulated by the auxin [114,115]. In addition, Li et al. [116] demonstrated that the fusion of the SRDX transcriptional repressor motif in the C-terminal of OsPIL11 and OsPIL16 caused constitutively photomorphogenic phenotypes with short coleoptiles and open leaf blades in darkness. Nevertheless, OsPIL16 was able to bind to the N-box region of the *OsDREB1B* promoter, and it was substantially induced by cold stress in a *phyB* mutant [117]. In peach, PpPIF8 can interact with PpDELLA2 through an unknown motif, and the overexpression of *PpPIF8* in *Arabidopsis* promoted increases in the plant height and branch numbers [118]. Taken together, cross talk may exist among the grain development, light signaling and abiotic stress response that are meditated by *PIL* or *PIF* genes. These processes that connect the signal transfer to the molecular gene expression indicate the biochemical mechanisms of photomorphogenesis in plants (Figure 2).

Phytohormones, such as JA and abscisic acid (ABA), regulate plant growth, development, and defense processes. JA triggers the degradation of the protein jasmonate ZIM-domain (JAZ) by 26S protease, which induces the activation of multiple downstream JA-mediated responsive genes [119,120,121,122]. Some bHLH transcription factors are involved in the hormone signaling pathways in plants. *RERJ1*, which is a rice JA-responsive gene in subfamily III(a+c), was up-regulated via exposure to wounding or drought stress [123]. Interestingly, RERJ1 also interacted with OsMYC2 to mediate the defense processes against herbivory and bacterial infection through JA signaling [124]. The *DPF* in rice was also induced by JA, and as well as response to blast fungus infection, copper chloride, and UV light [104]. Additionally, JA-regulated *OsMYC2* induced the expressions of insect defense-related genes, and it simultaneously activated some of the biosynthetic pathways for the defense-related metabolites in rice, which was proven in a knockdown *osmyc2RNAi* plant (Figure 2) [124]. In *Artemisia annua*, AaMYC2-Like, the methyl jasmonate (MeJA) responsive transcription factor, played a prominent role in regulating the artemisinin biosynthetic pathway, as researchers confirmed through the transient overexpression of *AaMYC2-Like* in the leaves [125]. Furthermore, two MYC-type bHLH transcription factors from *A. annua*, AabHLH2 and AabHLH3 act as transcription repressors and functional redundantly to regulate artemisinin biosynthesis [126]. ABA plays a central role in a variety of physiological processes, and it is also a key abiotic stress-related hormone that responds to various environmental stresses in plants, such as cold, drought, and salt. Several bHLH transcription factors in subfamily IIIb are involved in abiotic stresses via the ABA signaling pathway. A group of *PebHLH* genes in moso bamboo (*Phyllostachys edulis*) possess various *cis*-elements for ABA and JA in their promoters, which are up-regulated by biotic and abiotic stresses, ABA and MeJA stimuli [31].

Under specific conditions, multiple phytohormones and environmental factors are constantly cross talking to affect plant growth and development. Based on previous studies, light usually promotes cell expansion in plant growth, while ABA and JA are normally involved in the biotic and abiotic stress responses. Some bHLH transcription factors play important roles in signal transduction networks that are mediated by plant hormones. Phytohormones and environment-responsive bHLH transcription factors participate in various plant developmental processes by interacting with each other either cooperatively or antagonistically to modulate plant growth.

### 2.4. bHLH Transcription Factors Are Related to Plant Abiotic Stress and Iron Homeostasis

Plants have evolved to adapt to the stressful conditions that are unfavorable for their growth and development [127,128,129,130]. However, these adverse environmental conditions substantially affect the yields of crops such as rice [131,132,133]. Cold stress impacts plant productivity, especially during the flowering stage, which substantially lowers the probability of reproductive success. Cold stress signals can be perceived by putative sensors and induce the expressions of stress-responsive genes to modulate the cellular activities through the cytosolic Ca^2+^ levels [133]. *bHLH* genes that are related to plant abiotic stresses are mainly in subfamily IIIb, including *OsICE1* and *OsICE2*, which researchers have widely identified in recent years (Table 1).

In rice, *OsbHLH148* responds to the initial JA signal and regulates drought responsive genes, including *OsDREBs* and *OsJAZs*, endowing the rice with drought tolerance [134]. *BEAR1*, which is a *bHLH* gene, regulates the salt response, which was demonstrated by the salt-sensitive phenotypes of its knockdown or knockout transgenic rice [135]. The *osbhlh024* mutant (A91), with a nucleotide base deletion generated by the CRISPR/Cas9 strategy, increased the shoot weight, and produced high antioxidant activities under salt stress, which indicates that *OsbHLH024* might play a negative role in the salt stress response of rice [136]. In addition, *OsWIH2*, which is a drought induced *WIH* gene, was activated by OsbHLH130, and the overexpression of *OsbHLH130* resulted in substantially higher drought tolerance via its participation in cuticular wax biosynthesis, with reductions in the water loss rate and ROS accumulation (Figure 2) [137]. OsbHLH057 targeted the AATCA *cis*-element, in the promoter of *Os2H16*, a gene responding to fungal attack in rice, and overexpression of *OsbHLH057* enhanced rice disease resistance to fungus *Rhizoctonia solani* and drought tolerance [138]. In sorghum, a typical *bHLH* gene, *SbbHLH85*, plays a key role in the root development, and its overexpression increased Na^+^ absorption, which indicates that *SbbHLH85* might play a negative regulatory role in salt tolerance [87].

The ectopic expression of *CabHLH035* in pepper (*Capsicum annuum* L.) enhanced the salt tolerance in transgenic *Arabidopsis* [139]. The overexpression of *MxbHLH18* in apple increased iron and high-salinity stress tolerances in *Arabidopsis* [140]. Zuo et al. [34] reported the genome-wide identification of the *bHLH* family genes in *Zoysia japonica*. The expressions of *ZjbHLH62/ZjICE2*, *ZjbHLH67*, *ZjbHLH76/ZjICE1*, *ZjbHLH88*, *ZjbHLH97*, and *ZjbHLH120* in subfamily IIIb were affected by cold, salt, dehydration, and/or ABA [34]. Furthermore, the overexpression of *ZjICE1* and *ZjICE2* endowed transgenic *Arabidopsis* and *Z. japonica* plants with abiotic stress tolerance via the activation of the *DREB/CBF* regulon, and they also enhanced ROS scavenging [141,142]. The overexpression of *MfbHLH38*, which is a *bHLH* gene of *Myrothamnus flabellifolia*, increased the tolerance to drought and salt stresses in transgenic *Arabidopsis* [143]. *HbICE2*, which is a novel *ICE-like* gene in the rubber tree (*Hevea brasiliensis*), is involved in JA-mediated cold tolerance and its overexpression enhanced the cold resistance in transgenic *Arabidopsis* [144]. The ThbHLH1 (subfamily XI) plays an important role in stress signaling pathways and induces the expressions of stress-related genes [145]. The bHLH transcription factors regulate a wide range of plant growth and stress response signaling pathways, and some of them share homeopathic pathways for plant survival under unfavorable or stressful conditions.

Metal deficiency (iron) substantially affects many physiological processes of plants, such as photosynthesis and respiration. Both low and high concentrations of iron greatly affect the growth of plants. Plants have developed regulatory systems to control their iron uptake and maintain their Fe homeostasis, and bHLH transcription factors play important roles in this process [146]. In rice, the *Iron-related transcription factor 3* (*OsIRO3*) in subfamily IVb is upregulated by environmental stress, and OsIRO3 is characterized as a negative regulator. The overexpression of *OsIRO3* reduced the gene expression in the process of Fe chelator biosynthesis that included *OsIRO2* [147]. Wang et al. [148] demonstrated that OsIRO3 formed both homodimers and heterodimers with the subgroup IVb bHLH transcription factor OsbHLH06 that is involved in the transcriptional repression processes in complex networks. Furthermore, researchers have also identified upstream positive bHLH regulators of *OsIRO3* in iron homeostasis, such as Photochemical Reflectance Index 1 (OsPRI1), OsPRI2/OsbHLH58 and OsPRI3/OsbHLH59, which are in subfamily IVc (Figure 2) [149,150]. *OsbHLH133*, in the subfamily VIIIc(2), plays an important role in the Fe-deficiency signaling network under Fe-deficient conditions in rice [151]. We present the results for other species, such as soybean, and apple, in Table 1. Manganese (Mn) is also an essential micro-nutrient that acts as a cofactor in the redox reactions in photosynthesis. *ZmbHLH105* confers improved Mn tolerance on the transgenic tobacco by repressing the expressions of the Mn/Fe-regulated transporter genes to reduce the Mn acclamation [152].

In summary, plant growth is strictly controlled by intricate regulation mechanisms, and the *bHLH* family genes always function with many other proteins to allow plants to perform specific developmental processes at suitable times, and to increase their chances of survival under unfavorable environments. Various research studies have demonstrated that bHLH transcription factors play important roles in a broad range of plant growth and developmental processes via crosstalk. For example, *RERJ1* mediates the defense processes against herbivory and bacterial infection through JA signaling [124], *OsPIL11* and *OsPIL16* are involved in photomorphogenic developmental processes that are affected by light [116], and *OsbHLH148* regulates drought stress in rice, which also responds to JA treatment [134]. The establishment of this mechanism is complicated, and systematic investigations into the *bHLH* genes in plants will facilitate their use for crop improvement programs.

**Table 1 ijms-24-01419-t001:** Functional characterization of basic helix-loop-helix (bHLH) proteins.

Subfamily	Gene Name	Gene Accession	Contribution	Reference
II				
	*ClATM1*	Cla010576	Anther development	[73]
	*OsEAT1*	Os04g0599300	Anther development	[68]
	*OsTIP2*	Os01g0293100	Anther development	[67]
	*SlMS10*	Solyc02g079810	Pollen and tapetum development	[75]
	*SlMS32*	Solyc01g081100	Pollen and tapetum development	[74]
III(a+c)				
	*CabHLH035*	LOC107866727	Salt	[139]
	*OsRERJ*	Os04t0301500	JA signaling	[123]
	*PebHLH35*	AIG53906	Drought	[153]
IIIb				
	*BjICE53*	HQ857208	Chilling (4 °C)	[154]
	*HbICE2*	AOO76749	Freezing (−8 °C)	[144]
	*OsICE1*	Os11g0523700	Drought	[155]
	*OsICE2*	Os01g0928000	Freezing (−6 °C)	[156]
	*ZjICE1*	QBQ01909	Freezing (−6 °C), drought, salt	[141]
	*ZjICE2*	QFQ50795	Cold, drought, salt	[142]
III(d+e)				
	*FtbHLH2*	KT737455	Chilling (4 °C)	[157]
	*OsMYC2*	Os10g0575000	Insect defense	[158]
IIIf				
	*DvIVS*	BAJ33515	Anthocyanin biosynthesis	[159]
	*OsS1*	Os04t0557500	Anthocyanin biosynthesis; JA signaling	[98]
	*PabHLH3*	KP126521	Phenylpropanoid metabolism biosynthesis	[103]
	*PabHLH33*	KP126523	Anthocyanin biosynthesis	[103]
	*PpbHLH3*	ppa002884m	Anthocyanin biosynthesis	[102]
	*StbHLH1*	JX848660	Phenylpropanoid biosynthesis	[160]
IVb				
	*OsbHLH062*	Os07g0628500	Iron homeostasis	[147]
	*OsIRO3*	Os03g0379300	Iron homeostasis	[147]
IVc				
	*OsPRI1*	Os08g0138500	Iron homeostasis	[149]
	*OsPRI2*	Os05g0455400	Iron homeostasis	[150]
	*OsPRI3*	Os02g0116600	Iron homeostasis	[150]
	*OsbHLH057*	Os07g0543000	Disease resistance, drought	[138]
	*ZmbHLH105*	AIB05526	Mn homeostasis	[152]
IVd				
	*OsDPF*	XM_015775745	Diterpenoid phytoalexin biosynthetic	[104]
	*OsbHLH024*	Os01g0575200	Salt	[136]
	*OsbHLH148*	NM_123731	Drought; JA signaling	[134]
	*SlAH*	KR076778	Anthocyanin biosynthesis	[100]
Va				
	*MfbHLH38*	QNN83755	Salt	[143]
Vb				
	*OsbHLH035*	Os01g0159800	Anther development	[71]
VII(a+b)				
	*OsbHLH107*	Os02g0805250	Grain development	[79]
	*OsPIL11*	Os12g0610200	light signaling	[116]
	*OsPIL15*	Os01g0286100	Grain development; light signaling	[78]
	*OsPIL16*	Os05g0139100	Grain development; light signaling	[77]
VIIIb				
	*ZmbHLH121*	GRMZM5G868618	Kernel development	[80]
VIIIc(1)				
	*BdRSL1*	XM-003565193	Root hair development	[86]
	*BdRSL2*	KQK00978	Root hair development	[86]
	*BdRSL3*	XP-010229851	Root hair development	[86]
	*OsRSL1*	NP-001047894	Root hair development	[85]
	*OsRSL2*	BAF03719	Root hair development	[85]
	*OsRSL3*	BAD46515	Root hair development	[85]
VIIIc(2)				
	*OsbHLH133*	Os12g0508500	Iron distribution	[151]
	*SbbHLH85*	SORBI_3008G147800	Root hair development	[87]
X				
	*AaMYC2-Like*	MH820174	JA signaling and Artemisinin biosynthesis	[125]
	*OsbHLH130*	Os09g0487900	Drought	[137]
XI				
	*ThbHLH1*	KM101094	Salt, osmotic stress	[145]
Orphans				
	*SlAR*	Solyc12g098620	Carotenoid biosynthesis	[105]
	*LoUDT1*	MW357612	Anther development	[72]
	*OsUDT1*	Os07g0549600	Anther development	[69]

Note: JA indicates Jasmonate. The abbreviations of the gene name from different species are showed as below: *Aa* refers to *Artemisia annua* L.; *Bd* refers to *Brachypodium distachyon*; *Bj* refers to *Brassica juncea*; *Ca* refers to *Capsicum annuum* L.; *Cl* refers to *Citrullus lanatus* L.; *Dv* refers to *Dahlia variabilis*; *Ft* refers to *Fagopyrum tataricum*; *Hb* refers to *Hevea brasiliensis*; *Lo* refers to *Lilium oriental*; *Mf* refers to *Myrothamnus flabellifolia*; *Os* refers to *Oryza sativa*; *Pa* refers to *Prunus avium* L.; *Pe* refers to *Populus euphratica*; *Pp* refers to *Prunus persica*; *Sb* refers to *Sorghum bicolor*; *Sl* refers to *Solanum lycopersicum* L.; *St* refers to *Solanum tuberosum*; *Th* refers to *Tamarix hispida*; *Zm* refers to *Zea mays*; *Zj* refers to *Zoysia japonica*.

## 3. Conclusions and Perspectives

Genetically modified organisms (GMO) have been widely developed using genetic engineering techniques to alter their original characteristics, such as flower color and shape, or tolerance for biotic and abiotic stresses. Transcription factors regulate different routes by modulating the respective downstream target genes, which can involve multiple plant physiological processes. Therefore, transcription factors are likely better targets for genetic engineering due to their broader regulatory capacities compared with other proteins. The identification of the bHLH transcription factors in different pathways contributes to our comprehensive understanding of the various aspects of plant morphogenesis and adaptation to extreme environments. In *Arabidopsis*, excellent reviews have covered the roles of bHLH transcription factors from different aspects, which include how bHLH transcription factors mediate *Arabidopsis* growth and development, such as flowering time control, seed dormancy and germination, and cell fate determination, how they function in environmental responses, such as iron homeostasis regulation, and abiotic stress responses, and how they respond to light and phytohormones [65]. Some bHLH transcription factors play crucial roles in many aspects in the crosstalk pathway, such as *BEEs* responding to brassinosteroids to regulate normal gynoecium development [17]. In this review, based on the functions of bHLH transcription factors, we classify the bHLH family that from important economic crops into four major categories (plant growth and development; metabolism biosynthesis; plant signaling, and plant abiotic stress), and we demonstrate that the bHLH family transcription factors play vital roles in plant growth, development, and adaptation to environmental stresses.

bHLH transcription factors regulate several pathways, and bHLH transcription factors in the same subfamily can make different contributions within plants. For example, subfamily VIIIc is mainly involved in root hair development, and *OsbHLH133* in subfamily VIIIc(2) plays an important role in the Fe-deficiency signaling network, which raises the question as to whether there is a relationship between root hair development and the Fe-deficiency signaling pathways. Whether *OsbHLH133* has any potential functions in root hair development is still unknown and poorly understood. Thus, bHLH members in the same subfamily may perform similar functions, but not necessarily. Our current knowledge of the bHLH transcription factors from different species mainly includes gene identification, and researchers have performed most of the functional characterizations in model plants (*Arabidopsis* or rice). Thus, the knowledge of the mechanisms that underlies these differences is still limited. The bHLH transcription factors from many economically important crops are not well characterized. In this review, we summarize the current research on the bHLH function in rice and other economical plants through an evolutionary classification, not only providing a relatively complete overview of the bHLH transcription factor family, but also indicating the potential functions of the bHLH transcription factors in different subfamilies. This review enriches our understanding of the bHLH family in rice, and it provides new insights into the bHLH transcription factor regulation in various biological processes.

Using genetic engineering tools to improve agronomic traits of crops is an efficient and rapid way for plant molecular breeding. Some bHLH transcription factors have been engineered in transgenic plants to increase plant tolerance, such as drought [134]. In addition, the regulatory capacity of transcription factor genes makes them better targets for genetic engineering, and they have high potential to produce broader responses compared with other proteins. In future studies, researchers should elucidate the mechanism of bHLH protein-regulated plant growth and development via the coordination of multiple pathways, such as phytohormones and environmental factors. In addition, the creation of genetically modified mutants and genetic manipulations of *bHLH* genes to identify additional regulation pathways in this complex process are also needed. Notably, most bHLH transcription factors that have been functionally investigated so far are evolutionary conserved, whereas the species-specific or tissue-specific bHLH transcription factors have been studied rarely. Thus, much more work is required to decipher the regulatory mechanisms of non-conserved bHLH transcription factors in plant growth, development, and response to biotic or abiotic stresses stimuli. Furthermore, new biotechnological tools will accelerate these complex processes, such as RNAi, virus-induced gene silencing (VIGS), the CRISPR/Cas9 gene editing system, single cell RNA sequencing, and omics analysis. Thus, the future seems bright with respect to the development of crops with improved agronomic traits that use bHLH transcription factors with a greater efficiency than ever before. Therefore, characterization of the bHLH transcription factors in various crops will enrich our understanding of their roles and evolution in plants, and provide new strategies for their genetic applications for plant engineering.

## Figures and Tables

**Figure 1 ijms-24-01419-f001:**
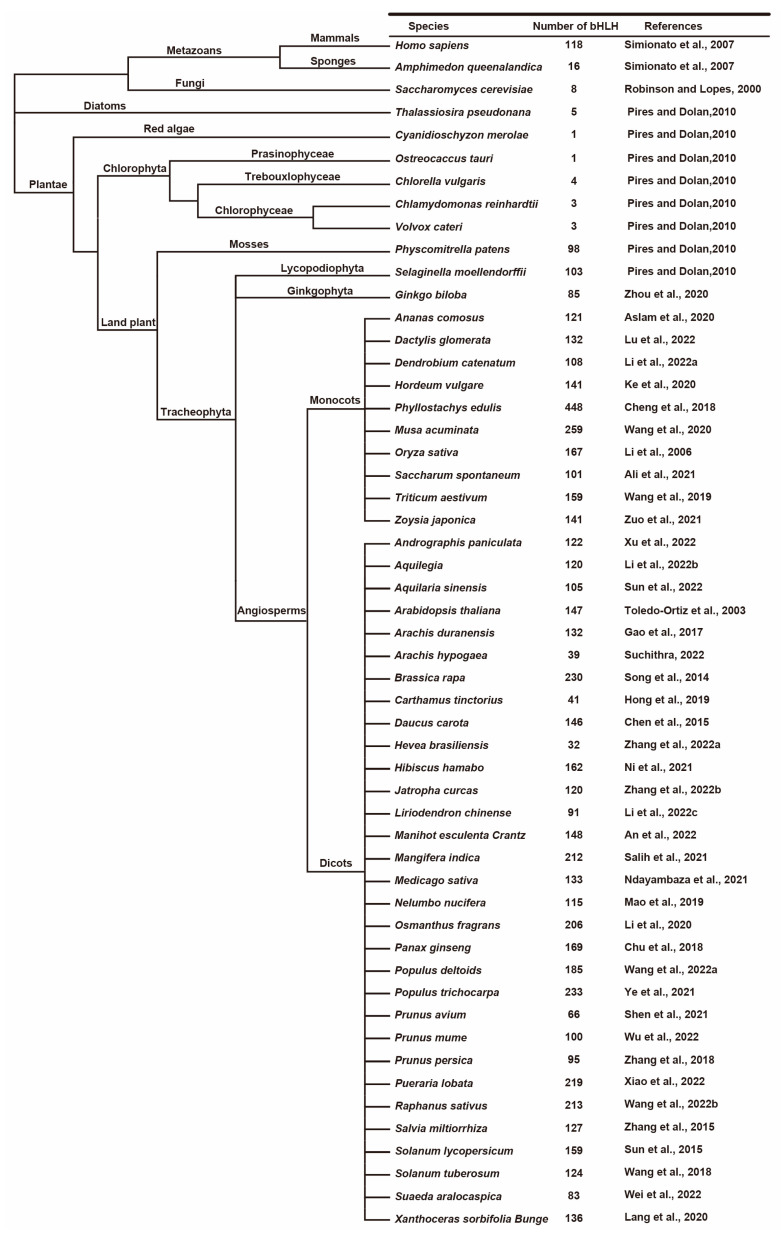
Phylogenetic analysis of the species based on genome-wide association study of the basic helix-loop-helix (bHLH) family. The total number of bHLH transcription factors identified from the genome of each species are indicated [5,22,23,24,25,26,27,28,29,30,31,32,33,34,35,36,37,38,39,40,41,42,43,44,45,46,47,48,49,50,51,52,53,54,55,56,57,58,59,60,61,62,63,64].

**Figure 2 ijms-24-01419-f002:**
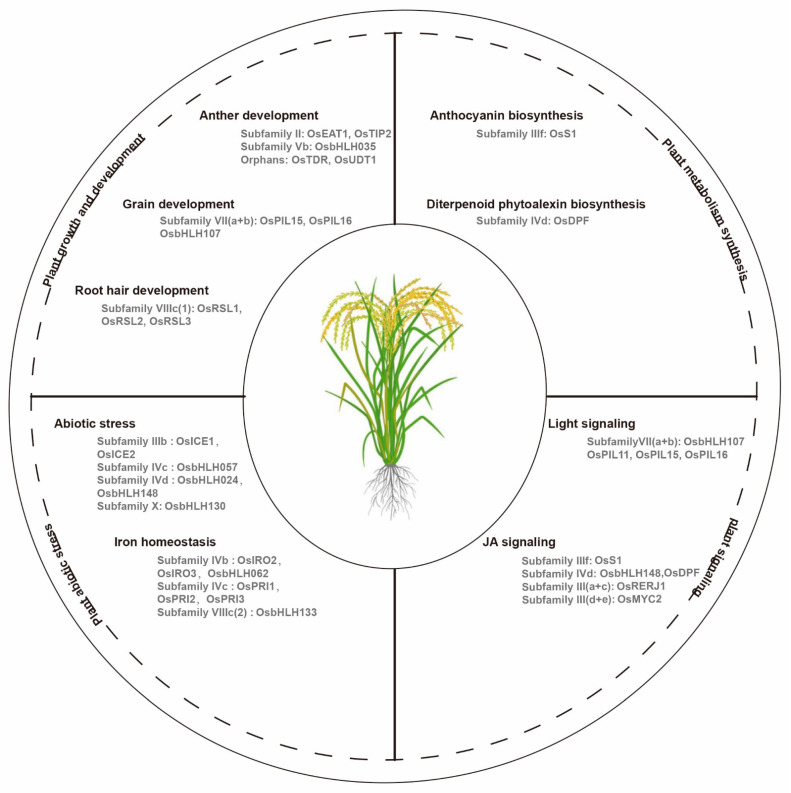
Functional classification of basic helix-loop-helix (bHLH) transcription factors in rice. *bHLH* genes are involved in various physiological processes for growth, metabolism, signaling, and adaptation to environmental stress.

## Data Availability

Data are available from the authors on request.

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
