# Peer review of "Basic Helix-Loop-Helix Transcription Factors: Regulators for Plant Growth Development and Abiotic Stress Responses"

_ijms, 2023, doi:10.3390/ijms24021419_

Round 1

Reviewer 1 Report

In this manuscript entitled “Basic helix-loop-helix transcription factors: Regulators for plant growth development and abiotic stress response” Zuo et al., discussed the role of bHLH transcription factors in growth and stress tolerance. The overall manuscript is well organized. I have some following concerns:

1-    The gene names must be in italics, please check the whole manuscript carefully.

2-    Abstract is too general; I will suggest that the authors improve it.

3-    In lines, 35-37: “Around 15 base pair α-helices regions in the C-terminus contains several hydrophobic amino acids, such as isoleucine (I), leucine (L), and valine (V), which promotes the formation of dimeric complexes”. Please add a reference.

4-    Authors mentioned that all the bHLH members were divided into six distinct groups (A-F). Out of these groups A and B both are bound to the E box, please confirm it.

5-    In Table 1, the font size is not consistent. Please check.

6-     bHLH also has role in biotic stress tolerance. I will suggest to the authors please add literature related to disease tolerance.

7-    Please provide some future solid recommendations.

Author Response

Point-by-point responses to Reviewer 1’s comments:

 # Reviewer 1

Comments and Suggestions for Authors

In this manuscript entitled “Basic helix-loop-helix transcription factors: Regulators for plant growth development and abiotic stress response” Zuo et al., discussed the role of bHLH transcription factors in growth and stress tolerance. The overall manuscript is well organized. I have some following concerns:

  1. The gene names must be in italics, please check the whole manuscript carefully.

Answer: Thanks for the reviewer’s suggestion. We have changed the gene names in italics throughout all the manuscript.  

  1. Abstract is too general; I will suggest that the authors improve it.

Answer: We have revised the Abstract and the revised are marked in blue.

  1. In lines, 35-37: “Around 15 base pair α-helices regions in the C-terminus contains several hydrophobic amino acids, such as isoleucine (I), leucine (L), and valine (V), which promotes the formation of dimeric complexes”. Please add a reference.

Answer: We added the Reference of the sentence‘Around 15 base pair α-helices regions in the C-terminus contains several hydrophobic amino acids, such as isoleucine (I), leucine (L), and valine (V), which promotes the formation of dimeric complexes [5]’in line 39 in blue.

  1. Authors mentioned that all the bHLH members were divided into six distinct groups (A-F). Out of these groups A and B both are bound to the E box, please confirm it.

Answer: Thanks for your important suggestion. We have changed the description in the sequences of E-box configuration that groups A and B bind to respectively in metazoans. They are marked in blue in line 44-46. Thank you very much again.

  1. In Table 1, the font size is not consistent. Please check.

Answer: We have changed Table 1 with same font size.

  1. bHLH also has role in biotic stress tolerance. I will suggest to the authors please add literature related to disease tolerance.

Answer: We have added some literatures that related to disease tolerance in blue in line70-71, line 299-302, line 445-447, line 634-635 in blue. We have made an English editing service that provided by the MDPI for this manuscript, and they are marked in green.

  1. Please provide some future solid recommendations.

Answer: We have added some solid recommendations for future studies in line 390-397 in blue.

Reviewer 2 Report

This is a very generalized review of the roles of the bHLH family of transcription factors in flowering plants. It is unclear what is the aim of this review. Arabidopsis s stated as an organism with extensively characterized bHLH family, however it seems the summary tends to lean in the direction of other plants. The review tends to be an incomplete literature list, rather than a focused analysis.

My overall suggestion to the authors would be the choose two or three biosynthetic or developmental or response pathways where bHLHs are known as central regulators and compare knowledge from different organisms, analyzing how information in one organism complements that from others or how organisms differ in regulating similar pathways. A good example is the iron response, see Gao et al (2019 Frontiers in Plant Sci) and Grillet and Shmidt (2019 New Phytol). An updated detailed comparison between the transcriptional regulation of Arabidopsis (Strategy I plant) and rice/maize (strategy II plants can be made and some important conclusions could be highlighted). Also, meta-analysis of transcriptomic data showed that bHLH transcription factors, such as PIF4 may be general coordinators of plant stress responses, possibly linking stress and development (Brumbarova and Ivanov 2019 iScience), which is a link that can be exploited by the authors to connect the diverse topics in the manuscript.

Further comments:

English: the manuscript needs extensive English editing. This includes Figure 1 (“mannals”), subheadings (chapter heading 2.1 wrongly suggests that all bHLHs are involved in growth and development; same is true for the rest of the subheadings)

Table 1: it is unclear based on what criteria the proteins resented in the table were chosen. There are a number of very well characterized bHLHs, for example from Arabidopsis thaliana and Oryza sativa that did not make it in this list.

There are inconsistencies between text and figures. Proteins that were mentioned in the text, for example tomato bHLHs related to anther development, were omitted from Figure 2.

The sentence on lines 156-157 lacks a reference.

Author Response

Point-by-point responses to Reviewer 2’s comments:

# Reviewer 2

Comments and Suggestions for Authors

This is a very generalized review of the roles of the bHLH family of transcription factors in flowering plants. It is unclear what is the aim of this review. Arabidopsis stated as an organism with extensively characterized bHLH family, however it seems the summary tends to lean in the direction of other plants. The review tends to be an incomplete literature list, rather than a focused analysis.

Answer: Thanks for the reviewer’s important suggestion. Functional characterized bHLH transcription factors have been summarized from different aspects in Arabidopsis thaliana including plant growth and development, stress response, biochemical functions, and the web of signaling networks [1-3]. In this review, we focused on the available knowledge about the functional characterized bHLH in some important economic crops, especially in rice.

References

  1. Hao, Y.; Zong, X.; Ren, P.; Qian, Y.; Fu, A. Basic Helix-Loop-Helix (bHLH) transcription factors regulate a wide range of functions in Arabidopsis. Int. J. Mol. Sci. 2021, 22, 7152.
  2. Gao, F.; Robe, K.; Gaymard, F.; Izquierdo, E.; Dubos, C. The transcriptional control of iron homeostasis in plants: a tale of bHLH transcription factors? Frontiers Plant Sci. 2019, 10, 6.
  3. Guo, J.; Sun, B.; He, H.; Zhang, Y.; Tian, H.; Wang, B. Current understanding of bHLH transcription factors in plant abiotic stress tolerance. Int. J. Mol. Sci. 2021, 22, 4921.

My overall suggestion to the authors would be the choose two or three biosynthetic or developmental or response pathways where bHLHs are known as central regulators and compare knowledge from different organisms, analyzing how information in one organism complements that from others or how organisms differ in regulating similar pathways. A good example is the iron response, see Gao et al (2019 Frontiers in Plant Sci) and Grillet and Shmidt (2019 New Phytol). An updated detailed comparison between the transcriptional regulation of Arabidopsis (Strategy I plant) and rice/maize (strategy II plants can be made and some important conclusions could be highlighted). Also, meta-analysis of transcriptomic data showed that bHLH transcription factors, such as PIF4 may be general coordinators of plant stress responses, possibly linking stress and development (Brumbarova and Ivanov 2019 iScience), which is a link that can be exploited by the authors to connect the diverse topics in the manuscript.

Answer: Thanks for your comment. In this review, we firstly collected the genome-wide identification of bHLH transcription factors from different species in recent years, and the Arabidopsis and rice also included in Figure 1, which provided helpful information for the functional studies of bHLH genes. Secondly, we summarized the available knowledge about the functionally characterized bHLH in some economic crops, and the bHLH transcription factors in rice have been showed in Figure 2. Here, although we have not covered all the information and recent trends in the published studies to bHLH family genes and their functional regulation pathways, we summarized them focusing on biological and physiological roles including plant growth and development, metabolism synthesis, signaling, and abiotic stress response at a view point of the application of those genes in crop biotechnology. Our work may provide genic resources and new strategies for genome engineering using the CRISPR/Cas9 system.

Further comments:

  1. English: the manuscript needs extensive English editing. This includes Figure 1 (“mannals”), subheadings (chapter heading 2.1 wrongly suggests that all bHLHs are involved in growth and development; same is true for the rest of the subheadings)

Answer: We have made an English editing service that provided by the MDPI for this manuscript. They were marked in green letter in the text. Thank you very much again.

  1. Table 1: it is unclear based on what criteria the proteins resented in the table were chosen. There are a number of very well characterized bHLHs, for example from Arabidopsis thaliana and Oryza sativa that did not make it in this list.

Answer: Thanks for the reviewer’s important suggestion. Functional characterized bHLH transcription factors from important economic crops have been showed in Table 1, the bHLH transcription factors from Arabidopsis were not included.

  1. There are inconsistencies between text and figures. Proteins that were mentioned in the text, for example tomato bHLHs related to anther development, were omitted from Figure 2. 

Answer: In Figure 2, we showed the functional classification of bHLH transcription factors from four parts only in rice, not in tomato.

  1. The sentence on lines 156-157 lacks a reference.

Answer: Thanks for your comment. We have added the reference of the sentence‘The rhizoid root hairs are essential for ion exchange, anchorage functions, and microbial interactions in the soil of land plants [39].’in line 162 in blue.

Reviewer 3 Report

The review titled “Basic helix-loop-helix transcription factors: Regulators for plant growth development and abiotic stress response” is a fantastic work for providing information to crop breeders. I have some minor suggestions to improve the paper for publication.

In the title, replace stress with stresses.

Line 11, replace stress with stresses.

Line 24, You use biotic stress. I think is it abiotic stress.

Line 54, use Pires and Dolan [ref. no.] rather than Pires and Dolan (2010). There is some other similar mistake like this, check throughout the manuscript.

Line 110, use Hao et al. [23] rather than Hao et al. (2021) [23]. There is some other similar mistake like this, check throughout the manuscript.

Line 64, replace show to showed.

Line 67, replace play with played.

Line 68, use and in between cold, salt stress.

Figure 1, use a high-resolution figure. Also for Figure 2.

Table 1, check the table and present properly.

Author Response

Point-by-point responses to Reviewer 3’s comments:

# Reviewer 3

Comments and Suggestions for Authors

The review titled “Basic helix-loop-helix transcription factors: Regulators for plant growth development and abiotic stress response” is a fantastic work for providing information to crop breeders. I have some minor suggestions to improve the paper for publication.

  1. In the title, replace stress with stresses.

Answer: Thanks for the reviewer’s suggestion. We have changed the abiotic stress response with abiotic stress responses in the title in blue.

  1. Line 11, replace stress with stresses.

Answer: We have changed the abiotic stress response with abiotic stress responses in line 11 in blue.

  1. Line 24, You use biotic stress. I think is it abiotic stress.

Answer: We have changed the biotic stress response with abiotic stresses response in line 25 in blue.

  1. Line 54, use Pires and Dolan [ref. no.] rather than Pires and Dolan (2010). There is some other similar mistake like this, check throughout the manuscript.

Answer: We have changed the references you pointed in line 57, 60, 117, 159, 195, 236, 308 in blue.

  1. Line 110, use Hao et al. [23] rather than Hao et al. (2021) [23]. There is some other similar mistake like this, check throughout the manuscript.

Answer: We have changed the references you pointed in line 117, 159, 195, 236, 308 in blue.

  1. Line 64, replace show to showed.

Answer: We have revised this sentence in line 66, 67 after English editing in blue.

  1. Line 67, replace play with played.

Answer: We have revised this sentence in line 69-72 after English editing in blue.

  1. Line 68, use and in between cold, salt stress.

Answer: We have added and between cold, salt stress in line 72 in blue.

  1. Figure 1, use a high-resolution figure. Also for Figure 2.

Answer: We have changed the figures with high-resolution, and we also provided the .svg figures. Thank you very much again. 

  1. Table 1, check the table and present properly.

Answer: We have changed Table 1 with same font size. We have made an English editing service that provided by the MDPI for this manuscript, and they are marked in green.

Round 2

Reviewer 1 Report

The authors have addressed all my concerns. I will suggest to the editor please accept it as it is.

Author Response

Thanks for Reviewer1’s approval, all comments you raised have much improved this article. Thank you very much again.

Reviewer 2 Report

While the English language has been edited and some improvements were introduced, I still find the review too general, unfocused and not bringing forward new ideas stemming from the critical detailed evaluation of published data.

Author Response

Thanks for Reviewer2’s comments, all suggestions you raised have much improved this article. Thank you very much again.